**communications** engineering

# Open-source interactive design platform for 3D-printed microfluidic devices
Yushen Zhang [ORCID], Mengchu Li, Tsun-Ming Tseng [ORCID] [✉] & Ulf Schlichtmann

Microfluidics and 3D printing offer exciting opportunities for the development of new technologies and applications in the fields of biology, chemistry, and medicine. However, the design of 3D-printed microfluidic devices remains a challenging and complex task, requiring specialized knowledge and expertise in fluid mechanics, 3D modeling, and 3D printing technology. Currently, there are very few tools helping engineers to do the labor-intensive process of designing microfluidic devices, let alone any tools that can help them design microfluidic devices for 3D printing. In this work, we introduce Flui3d, an interactive software platform for designing microfluidic devices for 3D printing. Flui3d offers a standard parameterized component library, support for multi-layer design, and the ability to design and configure microfluidic devices without the need for specialized knowledge. Flui3d incorporates a distinctive Design-for-Manufacturing (DFM) function, facilitating seamless fabrication of the designed microfluidic devices using commercial consumer-grade printers. We discuss the key features and benefits of Flui3d and demonstrate them by designing examples of microfluidic devices. We also discuss the design complexity and the potential applications of Flui3d.

Microfluidics is the science that relates to the behavior, precise control, and manipulation of fluids and particles in the range of tens to hundreds of microns[1]. It is a prosperous field that enables a wide range of biochemical and clinical applications, such as cancer screening, microphysiological systems engineering, high-throughput drug testing, and point-of-care diagnostics[2–4]. However, the development, from design to fabrication, of microfluidic devices is often complicated, time-consuming, and requires expensive equipment and sophisticated cleanroom facilities[5,6]. In recent years, three-dimensional printing (3D printing) has emerged as a promising alternative to traditional fabrication methods such as lithography and PDMS (polydimethylsiloxane) glass bonding. It enables rapid design iterations in the development phase and reduces institutional infrastructure costs, equipment installation and maintenance, and space requirements[7]. The utilization of 3D printing fabrication is increasingly widespread across various fields, allowing the fabrication of tiny structures, such as programmable microfluidic porous matrices[8] or conductive hydrogel structures[9]. With the advancement of 3D printing technology, not only single-layer microfluidic devices can be produced quickly and inexpensively using a 3D printer, but methods to create multilayer microfluidic devices are also emerging, shrinking the surface size of the device to another level[10,11]. Stereolithography (SLA) is a popular 3D printing technology for microfluidics fabrication. This method employs a light source to cure a photosensitive resin layer by layer. SLA yields 3D chips of higher precision and resolution than alternative printing technologies such as PolyJet or Fused Deposition Modeling (FDM)[12]. However, factors such as light penetration

depth can affect the quality of the printed device immensely, particularly in microfluidic devices with small features or multiple layers.

Despite the advantages of 3D printing for microfluidics, challenges remain in producing small and multilayer 3D-printed microfluidics due to these limitations. Not only does the light penetrate through transparent material, causing curing processes at unintended locations, but the printer characteristics and resin properties can also significantly impact the final result. While researchers have proposed solutions such as manipulating the 3D printer or using customized resin[13,14], these methods are not widely available or accessible. Nonetheless, the field of 3D printed microfluidic devices is constantly evolving and is gaining popularity and widespread use in various fields[15–17]. Besides, the recent proposal of an active control mechanism for 3D-printed microfluidics brings even more possibilities to the technology[18].

Designing microfluidic lab-on-a-chip systems is an arduous task requiring specialized knowledge in fluid dynamics, mechanical design, and manufacturing[19]. Engineers face substantial challenges during the labor-intensive process of designing microfluidic devices. Typical design iterations require the engineer to research architectures, manually design device layouts, and optimize manufacturing processes[20]. The task becomes more complicated when designing microfluidic devices for 3D printing fabrication and even more for multilayer devices. Not only do the engineers need to understand how to draw drafts in three-dimensional space using 3D capable modeling software or computer-aided design (CAD) tools like Solidworks and AutoCAD, but they also have to learn the skill of fabricating three-

Chair of Electronic Design Automation, Technical University of Munich, Munich, Germany. [✉]e-mail: tsun-ming.tseng@tum.de

dimensional models, putting more effort into adding another dimension to the design process. Although state-of-the-art 3D modeling software and CAD tools have achieved widespread adoption across a diverse range of industrial and academic fields, their inherently general-purpose nature falls short in addressing the highly specialized and intricate demands of designing microfluidic devices for 3D printing, resulting in a significantly less efficient design process. The necessity to further lower barriers to accessing 3D-printed microfluidics research persists.

In recent years, Electronic Design Automation (EDA) tools specifically for conventional, or in other words, 2D, microfluidics have advanced rapidly, handling most of the labor-intensive design tasks automatically. State-of-the-art microfluidics design automation tools, such as Cloud-Columba[21,22], 3DµF[23], Micado[24], or Fluigi[25], have immensely helped engineers in the design process of continuous-flow microfluidic devices. They can output 2D vector graphics or CAD scripts that can be used for conventional fabrication methods. However, none of the microfluidics design automation tools is applicable for designing microfluidic devices for 3D printing. Therefore, with the potential benefits of adopting domain-specific CAD tools, the need to develop design automation for 3D-printed microfluidics arises.

With that in mind, in this work, we introduce Flui3d—an interactive design platform for designing microfluidic devices for 3D printing fabrication that can dynamically optimize the design to achieve a manufacturing-ready output. One of the key features of Flui3d is its automated post-design processing utilizing our design-for-manufacture (DFM) function, which dynamically compensates for light penetration by adding additional height or space during the output stage to prevent complete curing of unintended areas, enabling the fabrication of small and multilayer microfluidic devices without the need for custom 3D printers or resins. With the integrated parametric microfluidic component library, users can quickly and easily create designs through simple mouse clicks. Besides planar microfluidic device structure design, taking advantage of 3D printing fabrication, Flui3d offers a multilayer co-design strategy allowing users to design even complex microfluidic devices for 3D printing in a straightforward way. Flui3d offers a design workflow that requires no prior knowledge of 3D modeling techniques from the users. Users simply place modules and components in a familiar 2D fashion, and Flui3d automatically generates the corresponding 3D design, producing manufacturing-ready files as the output.

Designs can be exported directly as STL (Standard Triangle Language) files, a standard format for 3D printing, or as SVG (Scalable Vector Graphics) vector graphics for other types of fabrication. Users can also save their designs in a human-readable JSON (JavaScript Object Notation) format that can not only be shared across the community, allowing others to quickly develop new devices by reusing shared designs but also be read by other programs capturing design and architectural information easily. In sum, our work provides the following contributions: (1) an intuitive design platform featuring a design mechanism tailored for 3D-printed microfluidics enabling rapid prototyping, (2) two compensation models for automated post-design processing enabling design-for-manufacturing for commercial consumer-grade 3D printers.

We demonstrate the high efficiency of Flui3d in designing microfluidic devices and components for 3D printing. We used seven devices from the literature and compared the design process with state-of-the-art design tools. We fabricated each design presented in this paper using a consumer-grade 3D printer. To achieve high-quality printing results, our DFM function was involved during the STL print file generation process. Our results demonstrate that even a consumer-grade 3D printer can efficiently print microfluidic devices designed using Flui3d. We believe that Flui3d will facilitate the spread of low-cost, 3D-printed microfluidics and make it more accessible to researchers and practitioners alike.

## Results
In contrast to traditional Electronic Design Automation software, Flui3d is an open-source web-based application providing a WYSIWYG (What You See Is What You Get) design interface. Without the need to install any

software, users can easily access the web-based design platform by opening the web address https://flui3d.org in the browser. This allows engineers to design their microfluidic designs for 3D printing using a browser on any computer or operating system. We made the source code of Flui3d available in "Code availability". In Table 1, we compare the distinct features available in Flui3d with those found in other commercial tools commonly used by researchers to design 3D-printed microfluidic devices, as well as state-of-the-art microfluidic design automation tools for 2D microfluidics.

Flui3d's design platform provides users with a simple user interface, as shown in Fig. 1, consisting of a design canvas with a *Task Toolbar* at the top, a *Design Toolbar* at the left, an *Information and Status Control Tool* at the right (visible by pressing the *Info/Status* button at the *Task Toolbar*) and a *Layer Control* at the bottom. The *Task Toolbar* provides general functions such as export, STL generation, etc. The *Design Toolbar* provides all design functions, such as placing a component, drawing a channel, etc.

### Flui3d design workflow
In this section, we demonstrate the design workflow of a three-layer, three-way mixing device for 3D printing. Figure 2 illustrates the complete design process of this device using Flui3d.

Prior to initiating the design process, users can specify the size of the microfluidic device and set default dimensional properties, such as the default component height and channel width, among others, using the *Property Inspector* in the *Information and Status Control Tool*, as illustrated in Fig. 2a.

In the first step, as depicted in Fig. 2b, the user selects the *Layer Control*, adds additional layers, and defines their positions as required. The user then selects a layer to design, as shown in Fig. 2c.

In the subsequent step, shown in Fig. 2d, e, the user opens the component library by selecting the component library button on the *Design Toolbar* and then parameterizes and creates the first mixer and places the mixer component onto the design canvas. The user then selects the next layer on the *Layer Control* and repeats the process to define and place other components. The user can also copy any components by selecting the components and using the shortcut Ctrl + C. The user can then paste them on any layer using the shortcut Ctrl + V.

In the next step, illustrated in Fig. 2f, the user adds ports by selecting the corresponding layer and using the port button on the *Design Toolbar*.

Following this, the user creates inter-layer connections (vias) between selected layers at the desired positions, as shown in Fig. 2g, and draws channels to connect each of the components, as illustrated in Fig. 2h. The user can then round the corners of the connection channels as desired, as shown in Fig. 2i.

Finally, the user can use the Design Rule Check (DRC) function to verify that the design complies with the specified rules, as shown in Fig. 2j. The design can then be generated as a 3D printable STL file. Depending on the user's printing requirements, Flui3d can also generate an optimized 3D model utilizing our DFM function that can be printed using SLA 3D printers, as demonstrated in Fig. 2k, l. The DFM function will be elaborated in detail in "Design-for-manufacturing". The entire process of constructing a three-layer 3D print-ready microfluidic design file using Flui3d takes less than 5 min, and the resulting design file is available in the Supplementary Data. Figure 3a depicts the individual layers of the final design.

We have fabricated this design using a consumer-grade DLP 3D printer, Anycubic Photon D2, using Miicraft BV-007A resin. Detailed printer and slicing settings are available in Supplementary Note 1. Figure 3b shows the fabricated three-layer three-way mixing device. In order to assess the versatility of Flui3d across diverse printing technologies and resins, we replicated the design using the Anycubic Photon D2 and two alternative resins, each possessing distinct characteristics. These resins include Anycubic Plant Based + Clear, an economical and non-professional hobby resin for home use, and Liqcreate Bio-Med Clear, a professional-grade bio-compatible resin. In addition, we extended our investigation by reproducing the design on another entry-level consumer-grade MSLA 3D printer, the Elegoo Mars 4, employing the aforementioned three resins. Comprehensive

**Table 1 | Comparison of distinct features involved in designing microfluidic devices for 3D printing of different design tools**

| Features | Design tools | | | |
|---|---|---|---|---|
| | SW/AC/IV/S3 | Columba | 3DµF | Flui3d |
| **MICROFLUIDIC DESIGN TECHNIQUES** | | | | |
| Parametric component design | ✓ | Limited | ✓ | ✓ |
| Procedural component generation | | ✓ | ✓ | ✓ |
| Multilayer design for 3D printing | ✓ | | | ✓ |
| Geometric constraint | ✓ | | | Limited |
| **MICROFLUIDIC DESIGN AUTOMATION** | | | | |
| Automatic channel-component snap | Limited | | | ✓ |
| Component replication | ✓ | | ✓ | ✓ |
| Standardized microfluidic component library for 3D printing | | | | ✓ |
| 3D printing manufacturing optimization | | | | ✓ |
| Design rule check | | | | ✓ |
| Design synthesis | | ✓ | | |
| **GENERAL** | | | | |
| Custom 2D geometry creation tools | ✓ | | | ✓ |
| Custom 3D geometry creation tools | ✓ | | | ✓ |
| 3D printable STL file output generation | ✓ | | | ✓ |
| SVG file output generation | ✓ | ✓ | ✓ | ✓ |
| Fluidic simulation | ✓* | | | |
| Freeware | | ✓ | ✓ | ✓ |
| Open source | | | ✓ | ✓ |

*SW* SolidWorks, *AC* AutoCAD, *IV* Autodesk Inventor, *S3* Shapr3D.

*AutoCAD, Inventor, and SolidWorks offer limited simulation capabilities. COMSOL Multiphysics offers a solution called LiveLink, facilitating the integration of AutoCAD, Inventor, and SolidWorks designs. Notably, Shapr3D does not support any of these simulation options.

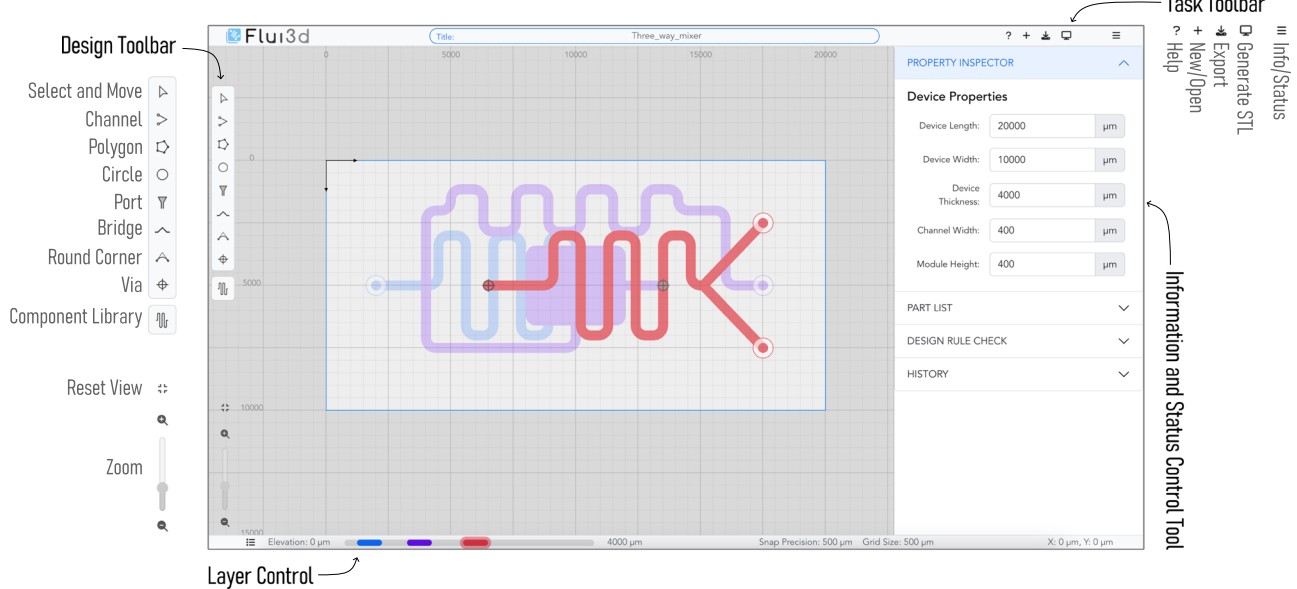

**Fig. 1 | Flui3d's user interface with a full-screen design canvas at the center.** The design canvas is surrounded by a Task Toolbar at the top, a Design Toolbar at the left, an Information and Status Control Tool at the right, and a Layer Control at the bottom.

insights into the outcomes, as well as detailed information on the printer and slicing settings, along with individual DFM settings, are provided in Supplementary Note 2.

**Microfluidics from literature with Flui3d**

In this section, we demonstrate the capabilities of Flui3d by replicating microfluidic devices from the literature. To illustrate the benefits of Flui3d, we select a representative set of microfluidic devices for 3D printing from the literature, including simple and complex devices, with a range of applications in different fields, such as biology, chemistry, and medicine. We also include another non-3D-printed microfluidic device that can be recreated with Flui3d for 3D printing to demonstrate the possibility of replicating conventional planar (flow-layer-only) PDMS-based devices for 3D printing purposes. Figure 4a–g shows the selected

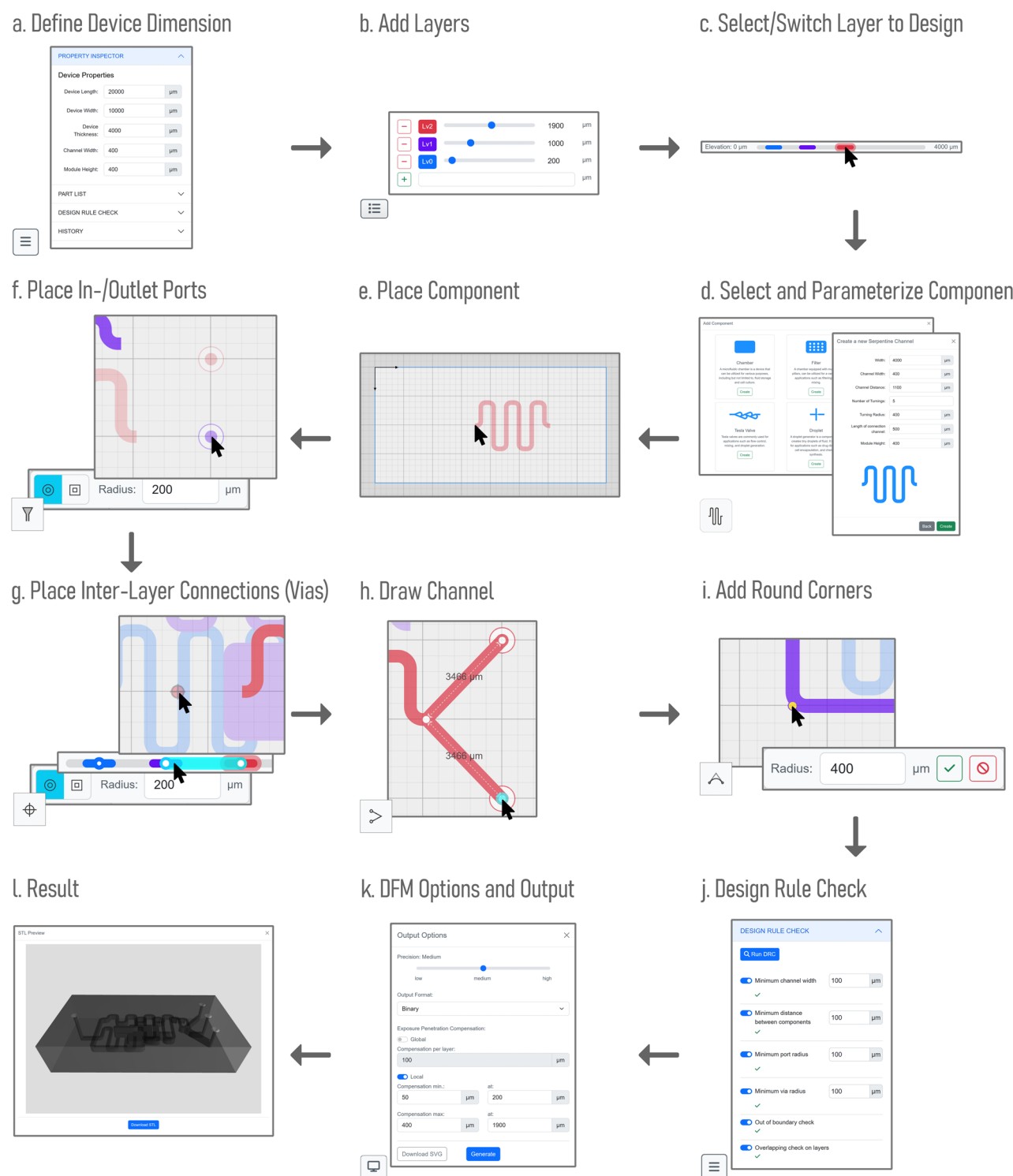

**Fig. 2 | Flui3d facilitates the design and fabrication of a three-layer, three-way microfluidic mixing device for 3D printing. a–l** illustrate the steps taken to design the device. The icon in the lower left corner indicates which button on the interface should be clicked to activate the function/operation shown.

designs that have been replicated using Flui3d: (a) Protein Immunoassays (orig. 3D-printed, Kadimisetty et al.)[26], (b) Resistive Microfluidic Networks (orig. 3D-printed, Tsur and Shamir)[27], (c) Genotoxic Evaluation (orig. 3D-printed, Kadimisetty et al.)[28], (d) SARS-CoV-2 Antibody Detection (orig. 3D-printed, Yafia et al.)[15], (e) 3D Droplet Generator (orig. 3D-printed, Tsuda et al.)[29], (f) Active Flow Control (orig. 3D-printed, Zhang et al.)[18] and (g) Planar Droplet Generator (orig. PDMS, Tan et al.)[30].

Each of the publications we referred to only included pictures or print files and brief descriptions of some design parameters, making it difficult to recreate the devices from what was provided in the paper. Figure 4h–n shows the resulting 3D models of the designs. All replicated designs using Flui3d were successfully fabricated (Fig. 4o–u) with the aforementioned consumer-grade 3D printer and resin to demonstrate the capabilities of Flui3d being employed in real-world applications. Detailed printer and slicing settings are available in Supplementary

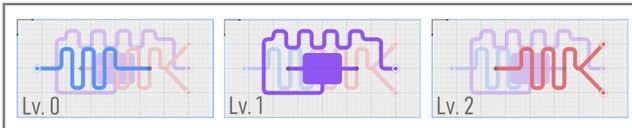

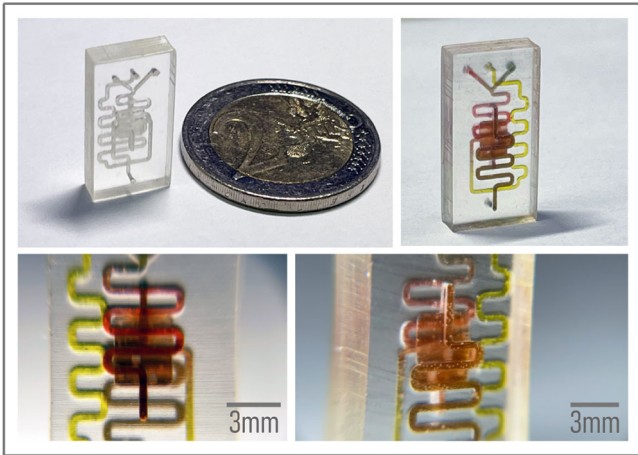

**Fig. 3 | Using Flui3d designed microfluidic device has been fabricated. a** depicts the individual layers, each accentuated by a distinct color. **b** shows the fabricated device using a consumer-grade 3D printer - Anycubic Photon D2. For scale, it is compared to a two-euro coin, and the different colors of food dye indicate the different layers.

Note 1. A design and printing suggestion is also provided in Supplementary Note 3.

## Multilayer microfluidics for 3D printing and miniaturization

Multilayer microfluidic devices are becoming increasingly popular due to the capabilities and versatility of 3D printing technology. These devices often require complex and interrelated components and can be challenging to design and fabricate using traditional manufacturing methods. The multilayer design feature of Flui3d not only enables users to create complex and functional microfluidic devices but also allows for easy and efficient modification and optimization of designs. By adding, removing, or modifying components and connections on different layers, users can quickly and easily refine their designs to achieve the desired performance and functionality.

In "Microfluidics from Literature with Flui3d", some of the designs that we have replicated using Flui3d were designed using Flui3d's multilayer design feature (devices (a), (b), (c), and (e) from the literature). Similarly, the three-layer, three-way mixing device that we used to demonstrate Flui3d's design workflow was multilayer designed.

Designing multilayer microfluidic devices for 3D printing poses a substantial challenge due to the need for high precision and accuracy in printing small features. This is particularly challenging because 3D printers may distort or deform overlapped small features during the printing process. To address this challenge, Flui3d incorporates the integrated DFM function that enables users to automatically optimize their design outputs to ensure successful printing, even when the features are small. More details can be found in "Design-for-manufacturing".

By utilizing the DFM function, we were able to reduce the size of some multilayer microfluidic devices from the literature that were originally designed to be larger and print them without any issues. For instance, devices (a) and (c) in "Microfluidics from Literature with Flui3d" were redesigned with a much smaller size factor, and their outputs were optimized using the local compensation method provided by our DFM function, which enabled us to print the output 3D models without any

modifications on the designs. The miniaturized designs were printed using the same consumer-grade 3D printer mentioned previously. Figure 5a, b displays the miniaturized designs and the corresponding fabricated devices, along with a side-by-side size comparison with the original-sized devices. The miniaturized designs are available in the Supplementary Data.

Further, to ensure successful printing, the print file output of the three-way mixer from "Flui3d Design Workflow" and the device (b)/Resistive Microfluidic Networks in "Microfluidics from Literature with Flui3d" were also optimized using the local compensation method provided by our DFM function. Detailed DFM settings applied to each design are available in Supplementary Note 4.

To illustrate the effectiveness of the DFM function, Fig. 5c depicts the printing of the two miniaturized designs, the three-way mixer design, and the Resistive Microfluidic Networks design without utilizing the DFM function, resulting in clogged features.

## Design complexity

We compare the design complexity with widely used conventional generic 3D modeling CAD tools—Shapr3D, Autodesk AutoCAD, Autodesk Inventor, and SolidWorks. The complexity of designing the same microfluidic device for 3D printing can vary widely depending on the design, so we use the total number of actions or operations (mouse clicks, key presses, or pen and touch inputs on CAD software for tablets) required for an average engineer to complete the design to estimate the complexity required in each case.

Designing a microfluidic device for 3D printing consists of four stages: base design, components design, connections design, and assembling. Each of the first three design stages consists of draft design and three-dimensionalization (3Dify). The first stage—base design—usually requires users to add a plane and draft the outline of the device body and extrude it to the desired size. The second and third stages—components design and connections design—usually require users to add a plane, then draft the design with desired shapes and extrude each part. The final step is required for some 3D modeling CAD tools to perform a subtraction or union of different previously extruded bodies. Thus, the total complexity is the sum of all complexities at different design stages: $Complexity_{total} = Complexity_{base} + \sum Complexity_{components} + \sum Complexity_{connections} + Complexity_{assembling}$.

Figure 6 provides a summary of the complexities involved in designing the seven microfluidic devices from literature, chosen in "Microfluidics from Literature with Flui3d", using various 3D modeling CAD software and Flui3d. A detailed table outlining the individual complexities at each stage can be found in Supplementary Note 5. To note is that since the design strategy may vary from engineer to engineer, the reported complexities serve only as an approximation.

As we can see from Fig. 6a, the design complexity of Flui3d is much lower than generic 3D modeling CAD tools. Figure 6b shows how often each individual component occurs in each design. One of the key advantages of Flui3d over generic 3D modeling CAD tools is its ability to simplify the design process and reduce the complexity of designing microfluidic devices. While generic 3D modeling CAD tools typically require an average complexity of $Complexity_{components} \in O(n)$ to design a component, Flui3d enables users to design a component with a complexity of $Complexity_{components} \in O(1)$ using its component library (Note: Using generic 3D modeling CAD tools: O(n), where n is the number of individual shapes or lines drawn to define the component. For example, a serpentine channel with 20 lines is O(20). Using Flui3d: O(1), as Flui3d allows users to select and place a predefined and parameterized component from its library with a single action. For example, a serpentine channel with any number of turns is always O(1)). This is because Flui3d includes a library of predefined and parameterized microfluidic components that can be easily configured and placed on the microfluidic device with the desired properties. In contrast, generic 3D modeling CAD tools do not have a specialized component library for microfluidic design, requiring users to manually draw and define each individual shape of a component, which can be time-consuming and error-prone. Especially when it comes to components with repeating shapes,

such as a serpentine channel, generic 3D modeling CAD tools require users to redraw (or copy-and-paste and reposition) the same shape, such as the "L" shape in the serpentine channel, multiple times, increasing the complexity and effort of the design process. Similarly, if a component needs to be resized, each shape must be modified individually. Overall, the difference in design complexity of a single component between Flui3d and generic 3D modeling CAD tools is substantial ($O(1) < O(n)$), and as the design process of the entire device is based on individual components, the total design

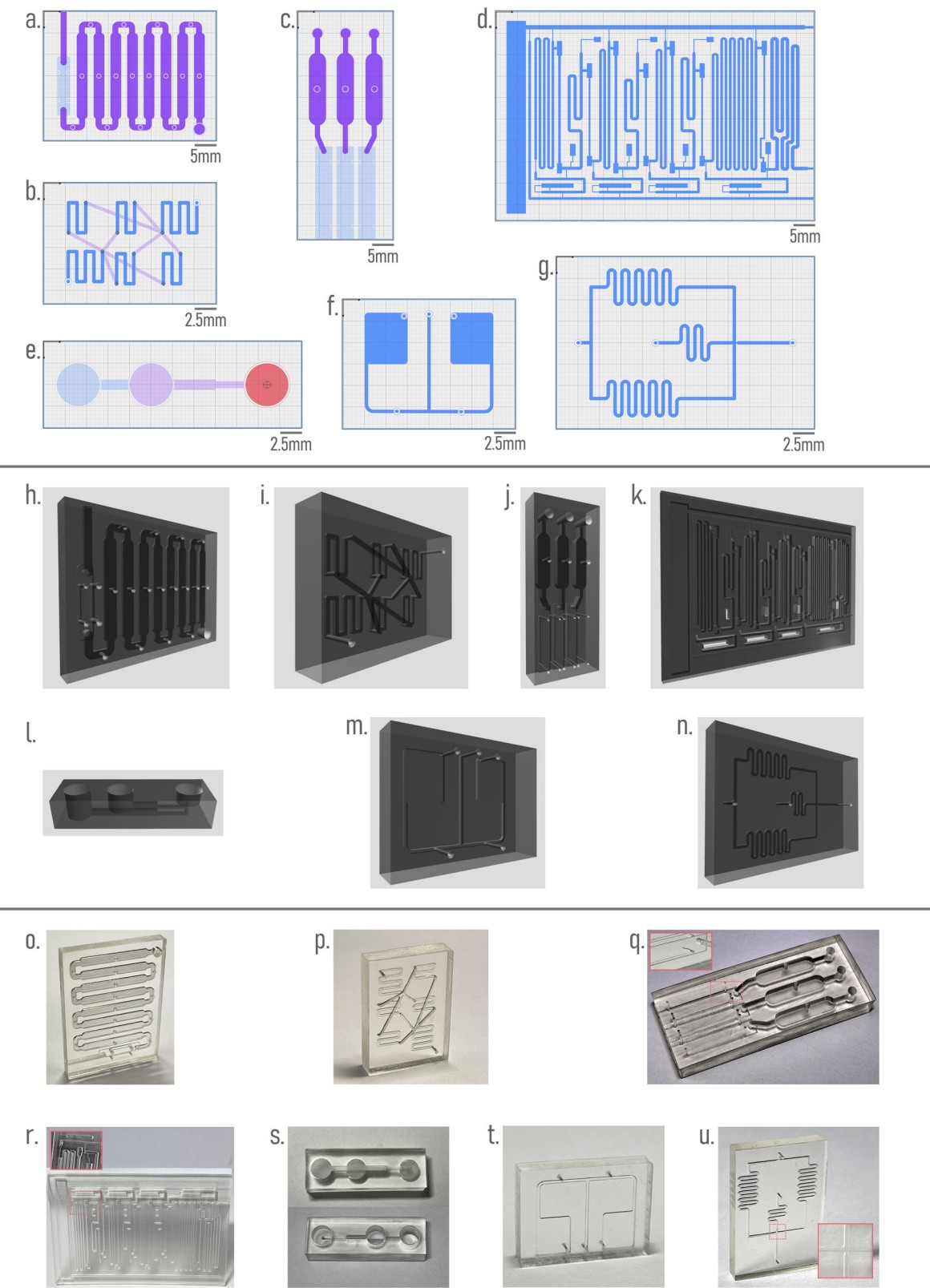

**Fig. 4 | Replicating designs from literature using Flui3d. a–g** depict the seven designs replicated using Flui3d, with each color representing a layer of the design. **h–n** correspond to the generated three-dimensional STL file models of designs (**a–g**) that can be printed using 3D printers. **o–u** show photos of 3D printing fabricated devices for each design.

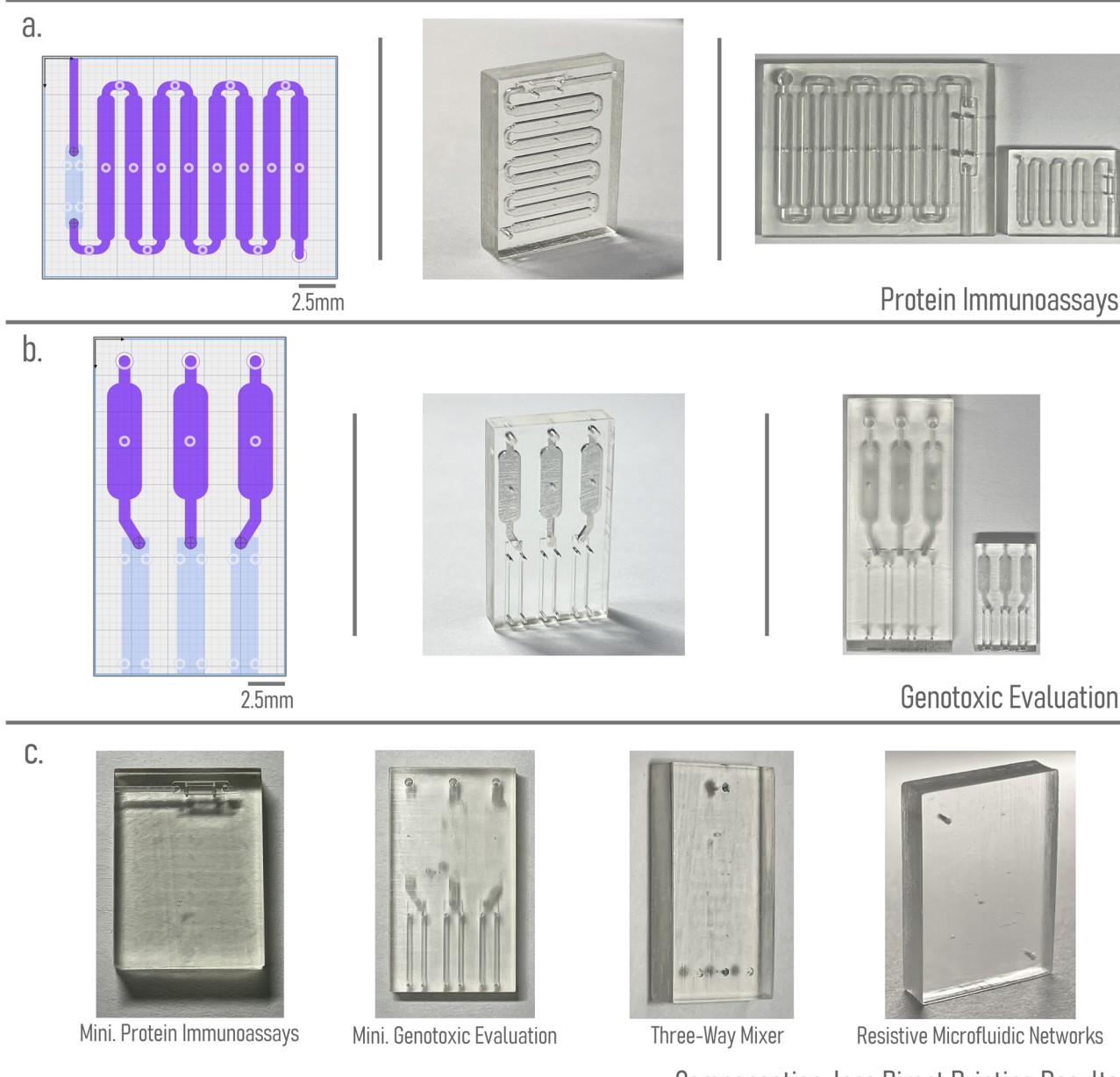

**Fig. 5 | Miniaturized microfluidic device designs from literature demonstrating the design-for-manufacturing feature of Flui3d. a** Protein immunoassay design reduced in size using Flui3d. **b** Genotoxic evaluation design reduced in size using Flui3d. For both (**a, b**), the left column shows the design in Flui3d, the center column shows the fabricated device using 3D printing, and the right column shows a side-by-side size comparison with the original device. **c** The 3D printing fabricated devices without using our DFM function.

complexity of a device is much lower than state-of-the-art generic 3D modeling CAD tools.

## Discussion

3D-printed microfluidic devices have become increasingly popular in recent years. However, designing microfluidic devices for 3D printing can be a challenging task, especially for researchers and engineers who are not familiar with the complex software tools required for this purpose. In this work, we introduced Flui3d, an open-source web-based platform that simplifies the design process for microfluidic devices.

One of the main advantages of Flui3d is its user-friendly interface, which allows users to design microfluidic devices using a browser on any computer or operating system. This eliminates the need to install any software, making Flui3d accessible to a wider range of users. In addition,

Flui3d offers a wide range of customizable design options, including a parameterized component library with several standardized components, allowing users to tailor the geometry and functionality of their microfluidic components to their specific needs and requirements.

To demonstrate the capabilities of Flui3d, we replicated several microfluidic devices from the literature, including simple and complex devices with a range of applications. The results showed that Flui3d was able to accurately replicate the designs of these devices, highlighting its potential as a reliable and efficient tool for microfluidic device design.

In addition to its user-friendly interface and customizable design options, Flui3d offers several advanced features that are not available in commercial tools commonly used by researchers to design 3D-printed microfluidic devices. For example, Flui3d includes a design-for-manufacturing (DFM) feature that helps users design microfluidic devices

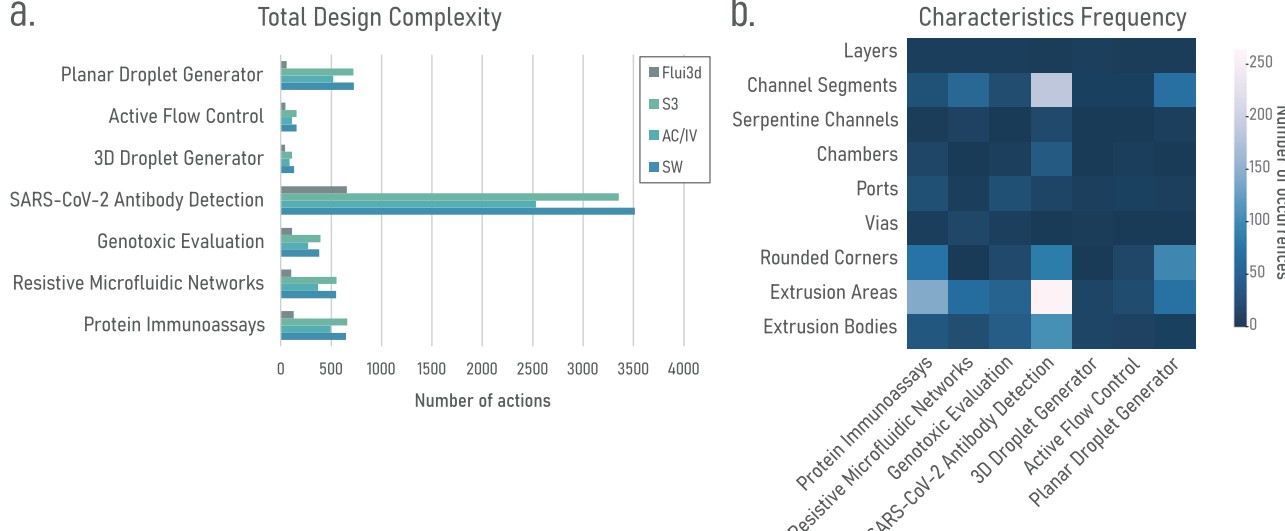

**Fig. 6 | Comparison of design complexities across different design tools and component frequencies of each design. a** Bar graph displays the design complexities of the seven designs from literature using different design tools. **b** Heatmap depicts the occurrence frequency of distinct components and features of each design, i.e., how many times each component and feature occurs in each design. S3 Shapr3D, AC AutoCAD, IV Autodesk Inventor, SW SolidWorks.

that can be easily and reliably manufactured using 3D printing. By using the DFM feature, we were able to reduce the size of some multilayer microfluidic devices from the literature that were originally designed to be larger. We printed these miniaturized designs without any problems.

Moreover, Flui3d allows users to specify several distance constraints compatible with the selected 3D printing technology to avoid design errors and manufacturing defects. It incorporates DFM principles by providing built-in design rules (DRC) that ensure the compatibility of the microfluidic design with the chosen manufacturing process. This is particularly important because one of the key challenges in the design of microfluidic devices is ensuring that the final design is manufacturable using 3D printing technology. Flui3d's DFM features help users design microfluidic devices that can be easily and reliably manufactured using 3D printing. By providing built-in design rules and constraints, Flui3d can automatically check the dimensions and tolerances of microfluidic channels and features and flag any potential problems or design rule violations.

Currently, Flui3d exhibits several limitations warranting future enhancements. A primary constraint pertains to the component library's scope, which currently lacks an extensive collection of high-quality and standardized microfluidic components. To address this deficiency, we actively encourage user participation in the collaborative expansion of its component repository, thereby enriching the design possibilities within Flui3d.

Furthermore, a second limitation manifests in the DRC and DFM functionalities within Flui3d. Presently, Flui3d offers a set of fundamental predefined design rules encompassing parameters such as the minimum distance between channels and components, minimum channel width, minimum port and via radii, out-of-boundary verification, and overlapping assessments. Divergent applications necessitate distinct design rule specifications, signifying the imperative need to extend the rule set to accommodate diverse microfluidic design contexts.

In addition, Flui3d's DFM function employs the Beer–Lambert law for calculating compensations. This process relies on approximations of various factors derived from user input. To enhance the precision of the compensation function, it is foreseeable that future iterations of Flui3d may necessitate a more refined and accurate calculation methodology. These enhancements collectively constitute vital prospects for the advancement of Flui3d as a comprehensive microfluidic design tool for 3D printing.

In summary, we introduced an open-source design automation platform—Flui3d—for 3D-printed microfluidics. Flui3d offers a user-friendly interface, integrated DFM, and customizable design options that can assist not only researchers and engineers but also students and members of the DIY community in designing microfluidic devices and components more efficiently. We have demonstrated that it is possible to fabricate microfluidic devices designed using Flui3d using a consumer-grade 3D printer, enabling rapid prototyping and low-cost microfluidics. We believe that Flui3d has made designing and fabricating microfluidic devices easier and more accessible than ever before.

## Methods

The design methodology of Flui3d is inspired by conventional generic 3D modeling software, CAD tools, and PCB (printed circuit board) editors. Flui3d combines the strengths and ease of each, which notably simplifies the design process of microfluidic devices for 3D printing. A description of software architecture and design is provided in Supplementary Methods. Subsequently, the following section describes the major features of Flui3d in detail.

### Standard parameterized component library

Redesigning established components in microfluidic contexts can be a time-consuming and laborious process. State-of-the-art PCB design software offers standard component libraries that contain many standardized component footprints for reuse purposes. In contrast to conventional electronic components, which are usually uniform in size, shape, and footprint within the same category, microfluidic components exhibit distinct characteristics. For instance, all 0603 resistors or capacitors, no matter what value they have, will have the same footprint on a PCB. For microfluidic components, even when they share fundamental properties like overall form and function, they can vary significantly in terms of their size, dimensions, certain shapes that are repeated, or the specific patterns or features they possess. For instance, small chamber vs. large chamber or 10-turn vs. 5-turn serpentine mixers. Consequently, to place the same type of microfluidic component with different properties, an engineer cannot simply replicate the component by copying and pasting. Instead, the engineer must redraw the entire component, adding to the effort required, particularly in the case of 3D printing microfluidic devices.

Thus, we designed the standard parameterized component library for 3D printing microfluidic devices, which is a collection of established, pre-designed, and customizable components that can be easily configured to meet the specific requirements of the larger design. This library

streamlines the creation of complex microfluidic designs, eliminating extensive manual drafting and calculation. The use of a parameterized component library can improve the consistency and reliability of the design, as the components have been pre-optimized for 3D printing and microfluidic performance. The parameterized component library standardizes the design process and reduces the time and effort needed to create new microfluidic devices.

As for now, we included several established microfluidic components for 3D printing in the library:

**Chamber.** Chambers are an essential component of microfluidic devices, as they provide the physical and chemical environment for the fluids to flow and interact and can be used to perform a wide range of functions, such as mixing, splitting, sorting, detecting, and reacting. We parameterized it with the following properties: width, length, height, and corner radius.

**Chamber with pillars.** A chamber with pillars is a chamber that has an array of pillars or posts inside, which can be used to control and manipulate the flow and behavior of fluids in the chamber. For example, the pillars can create vortices, jets, or eddies in the flow, which can mix, stir, or transport the fluids in the chamber; or can create barriers, traps, or channels in the flow, which can sort, filter, or separate the fluids in the chamber. We parameterized it with the following properties: width, length, height, corner radius, number of pillar rows, number of pillar columns, and pillar radius.

**Serpentine channel.** Serpentine channels are channels that have a serpentine or zigzag shape, which can be used to control and manipulate the flow and behavior of fluids in the channel. For example, serpentine channels can be used as mixing channels, where the fluids can be mixed and stirred by the serpentine shape of the channel, or as delay channels, where the flow speed of the fluid slows down physically. We parameterized it with the following properties: channel width, length, height, channel distance, number of turnings, and corner radius.

**Tesla valve.** A Tesla valve is a passive valve with no moving parts. The working principle of this valve is that forward flow experiences hydraulic resistance due to the looped shape of the conduit, while reverse flow experiences little to no resistance. Tesla valves have been used in various applications[31]. For example, Tesla valves are used in hydrogen fuel cell research, where the valve can be used as a decompression unit or as a micropump based on thermal cavitation. We parameterized it with the following properties: channel width, height, number of segment pairs, segment length, and segment width.

**Droplet generator.** A droplet generator is a component that can be used to produce droplets of fluids in a microscale or nanoscale environment. For example, a droplet generator can be used to encapsulate and protect a drug or a therapeutic agent inside a droplet. We parameterized it with the following properties: disperse channel width, continuous channel width, droplet channel width, droplet channel length, connection channel length, and height.

**Channel width transition.** A channel width transition is a microfluidic component that can be used to connect two channels with different widths. The channel width transition consists of a narrow segment with a tapered shape, which gradually widens or narrows from one channel to the other. The channel width transition can be used in a wide range of applications. For example, it can be used to connect a high-pressure channel with a low-pressure channel or to connect the main channel with side channels to enable the flow of the sample or the reagents into or out of the main channel. We parameterized it with the following properties: transition length, transition width left, transition width right, connection channel length, and height.

**Channel height transition.** Similar to the channel width transition, this component can be used to connect two channels with different heights. We parameterized it with the following properties: transition length, transition height left, transition height right, connection channel length, and height.

The standard parameterized component library in Flui3d can greatly enhance the capabilities and performance of microfluidic design for 3D printing. In the future, we plan to include additional components in the library. As open-source software, we also hope that it can facilitate collaboration and communication among researchers, scientists, and engineers in the field of microfluidics and can enable the development of new and exciting components that can be easily added to the library.

## Custom component design

Custom component design is an essential aspect of microfluidic device design, as it allows engineers to create unique and specialized components that are not available in the standard component library. However, custom component design can be challenging and time-consuming, as it requires a high level of technical expertise and precision.

Flui3d addresses this challenge by providing a simplified custom component design function that enables engineers to easily create and customize their own microfluidic components. With this function, engineers can quickly and easily draw simple shapes in solid, hollow, or stroke (channel) form, such as triangles, rectangles, and circles, using the tools provided on the Design Toolbar. These shapes can then be combined and modified to create more complex and specialized components, as illustrated in Fig. 7a, b. In addition, by combining the shapes with channels or standardized microfluidic components, users can create their own specified components rapidly.

It allows engineers to create a wide range of custom components that are tailored to their specific needs and requirements. For example, an engineer can create a custom microfluidic chamber with a specific pillar shape. This enables engineers to create microfluidic devices that are highly customized and optimized for their specific applications and goals.

However, the benefits of custom component design should extend beyond individual users. Users can download their custom designs' project files, and by sharing their custom designs with the microfluidic community, such as Metafluidics[32], users can contribute to the collective knowledge and expertise of microfluidic design. This enables other users to access and use these custom components in their own designs, increasing the range of available components and fostering collaboration and innovation.

## 3D microfluidics and multilayer design

The utilization of 3D printing technology confers a distinct advantage in its capacity to manufacture intricate and three-dimensional structures. The design of multilayer microfluidic devices presents a formidable challenge, demanding a profound comprehension of fluid dynamics and 3D printing technology. While generic 3D modeling software offers a range of tools and features for creating and editing 3D models, they may not be optimized for the design of multilayer microfluidic devices. This is primarily due to the intricate and interconnected nature of the components and features within microfluidic devices, requiring precise positioning and alignment across different layers. Designing multilayer microfluidic devices using generic 3D modeling software can be time-consuming and challenging, as it requires a high level of expertise and attention to detail. It is often tedious to visualize and identify the interrelationships and interconnections between different components and layers, especially in the context of intricate and densely configured designs. It can also be hard to accurately position and orient the components and features, as the software may not provide a good view of the layers or a clear delineation of the spatial dimensions of the components.

Flui3d presents a user-friendly way tailored to the design of multilayer 3D-printed microfluidic devices, thereby simplifying and streamlining the creation and specification of designs, along with the establishment of interlayer relationships and connections. To facilitate the design process, Flui3d includes a layer control tool at the bottom of the screen, which allows users

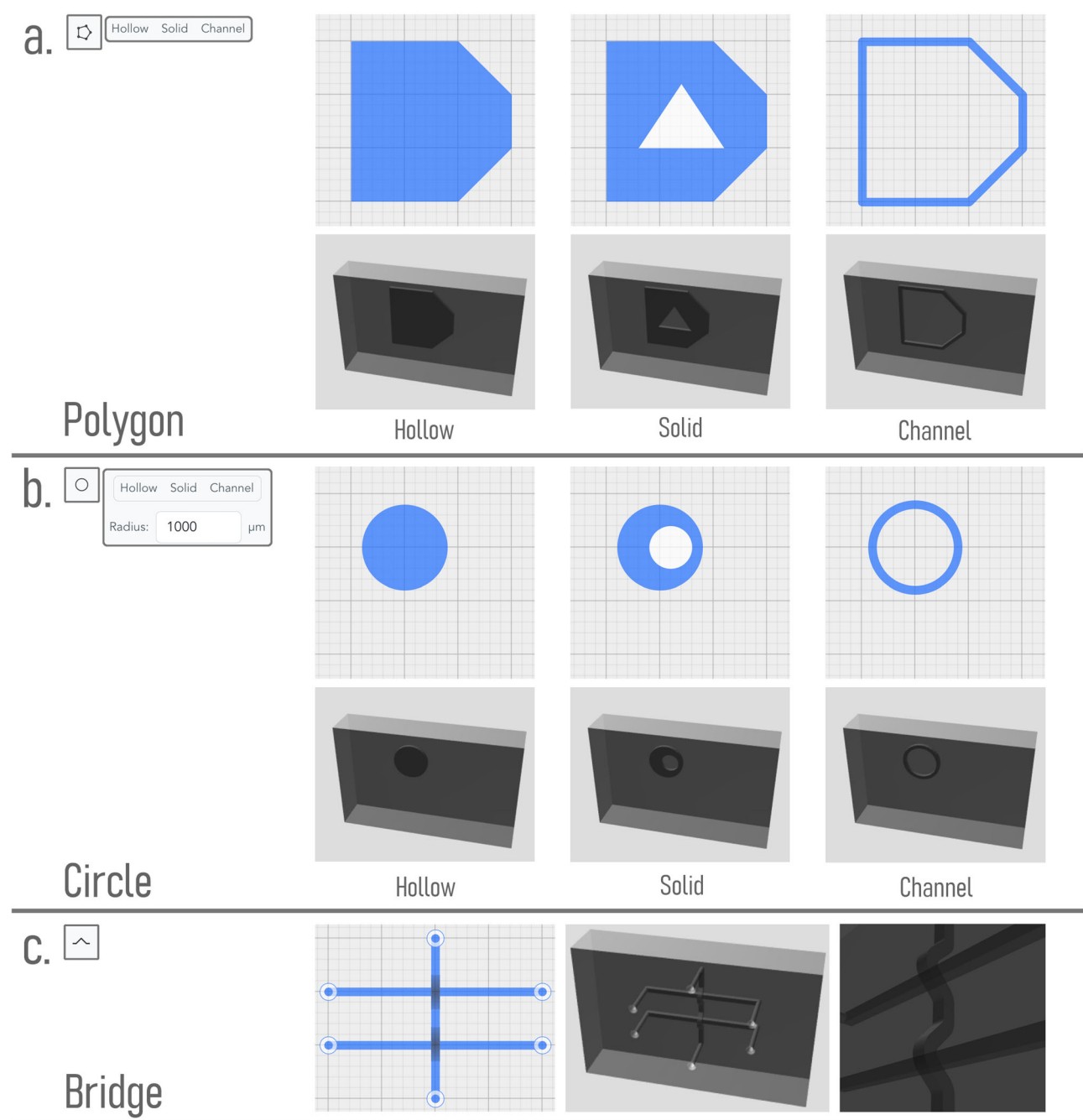

**Fig. 7 | Demonstration of special design features. a, b** Custom component design features of Flui3d. Users can create customized components by combining shapes using Flui3d's polygon and circle design tools. The designs are depicted at the top and the generated three-dimensional models are at the bottom. **c** The bridge function allows users to add a bridge to an existing channel, enabling them to avoid channel crossings, for example.

to easily add, remove, and change the layers. Each layer is automatically accentuated with a different color, which is displayed on a bar at the bottom of the screen, indicating the height of each layer (Fig. 2c). Users can select any layer to start designing the features on that layer. On the canvas, components and features at different layers are also indicated with different colors, so users can easily see where the features belong and on which layer they are currently working. When a layer is selected, all other layers' components and features are faded in color, so users can focus on the current layer design. An example is illustrated in Fig. 3a.

Leveraging the inherent advantages of 3D printing technology in the field of three-dimensional structural fabrication, we also present a feature in Flui3d—a bridge function. This function enables users to incorporate an arch bridge within their designs. The primary utility of this bridge function is to avoid channel crossing within the same layer of the printed structure. Figure 7c shows two bridges over two channels.

## Design-for-manufacturing

One of the key challenges in the design of microfluidic devices is ensuring that the final design is manufacturable using 3D printing technology. This is where design-for-manufacturing (DFM) comes into play. DFM is the process of designing products in a way that takes into account the manufacturing processes and capabilities in order to optimize the design for manufacturability, reliability, and cost.

Flui3d includes several DFM features that help users design microfluidic devices that can be easily and reliably manufactured using 3D printing. Like the state-of-the-art PCB design software, Flui3d allows users

**Fig. 8 | Comparison of a three-layer design output with and without compensation using different compensation strategies.** Four vertically cross-cut side views of STL models are shown with the same view angle. **a** shows the output of a design without using compensation. **b**, **c** show the output with local and global compensation, respectively. **d** shows the output with both local and global compensation. With local compensation involved, each feature layer will be compensated with gradually increasing height calculated automatically.

to specify several distance constraints compatible with the selected 3D printing technology to avoid design errors and manufacturing defects. It incorporates DFM principles by providing built-in design rules and constraints (DRC) that ensure the compatibility of the microfluidic design with the chosen manufacturing process (Fig. 2j). We have established a set of design rules and constraints that users can individually configure and activate. These include minimum channel width, minimum component distance, minimum port radius, checks for out-of-chip boundaries, and detection of overlapping objects on the same layer. For example, Flui3d can automatically check the dimensions and tolerances of the microfluidic channels and features and flag any potential issues or violations of the design rules. This helps users avoid common mistakes and pitfalls that can lead to failure or poor performance of the microfluidic device.

As mentioned before, currently, one of the most popular 3D printing technologies that is used for fabricating microfluidic devices is stereolithography (SLA) printing technology, which is an overall term for Laser SLA, Digital Light Processing (DLP) SLA and Mask-SLA (LCD 3D printing) technology. SLA printing technology uses a laser or other light source to cure a photosensitive resin layer by layer, creating a 3D object. In the context of microfluidics, this technology can be used to create complex microfluidic devices with high resolution and accuracy. Nevertheless, the quality of the printed device can be influenced by a number of factors, including the light penetration depth. The light penetration depth refers to the distance that the light can travel into the resin before it is fully absorbed. This distance is determined by multiple properties, such as the properties of the resin, the wavelength of the light used, the intensity of the light source, and ambient temperature and humidity. A common problem when using stereolithography to print transparent or semi-transparent unibody—microfluidic devices that all features are printed (enclosed) into the device in a single printing process, and no additional lamination or bonding process is required—or multilayer microfluidic devices is that because the resin is transparent, the light used to cure it can pass through several layers (print slices) and cure areas that are not intended to be solid. This can cause the resin to solidify in unintended areas, resulting in defects or flaws in the printed features[33], as presented in Fig. 5c.

Fabricating microfluidic devices with small features is particularly difficult, especially when it comes to multilayer devices. As demonstrated by examples in the literature, many 3D-printed microfluidic devices are large and could more accurately be classified as "mili-fluidic." Consequently, small and multilayer 3D-printed microfluidics are not commonly seen due to these challenges. We propose a way that is able to overcome this issue. By adding additional height or space to the designs, we could compensate for light penetration and prevent the complete curing of unintended areas.

As part of the DFM function, Flui3d supports the optimization of microfluidic devices for manufacturing and includes this unique feature for exposure compensation, which, at the output stage, can dynamically compensate the height of features in the design (local compensation) or add a user-specified blank exposure height (global compensation) to each feature layer in order to optimize the design for the SLA printing technology (Fig. 2k). Figure 8a–d illustrates the difference between output (a) without compensation, (b) with local compensation, (c) with global compensation, and (d) with both compensation methods.

For the local compensation, each feature layer will be compensated with a different height since the light intensity exposed to a layer in stereolithography printing falls exponentially with the distance between that layer and the light source (Eq. (1)), according to Beer–Lambert law.

$$I(z) = \alpha \cdot e^{-\beta z} \tag{1}$$

In general, the light absorbed by a layer is the total amount of light exposed to that layer. Thus, the total compensation $C$ of a layer at a height $Z$ is proportional to the received accumulated light intensity.

$$C(z) = \int I(z)dz$$
$$= \frac{\alpha \cdot e^{-\beta z}}{-\beta} + K \tag{2}$$

To enable automatically finding the compensation amount for each layer of a print, we allow users to input the desired minimum and maximum compensation values (represented as $C_{min}$ and $C_{max}$, respectively) and the corresponding height locations (represented as $Z_{min}$ and $Z_{max}$). This information is used to approximate the compensation needed for each feature layer of the print. For example, users may specify that they require a 100 μm compensation at a device height of 200 μm and a 500 μm compensation at a device height of 3500 μm.

The coefficients $\alpha$ and $\beta$ are used to describe the aforementioned factors, and they can vary depending on the printer, printing settings, resin properties, etc. These coefficients are calculated based on the information provided by the user. Additionally, the constant $K$ in Eq. (2) represents an offset of the compensation and should be positive. Like the coefficients, this offset can vary depending on the aforementioned factors.

To approximate the offset $K$, we use the information provided by the user, specifically $C_{min}$ and $C_{max}$, which represent the minimum and maximum compensation values. By summing these values ($C_{min} + C_{max}$), we can obtain an approximation of the offset. This approach enables users to achieve applicable compensation for their prints, even without knowing the exact relationship between the aforementioned factors, such as the printing technologies or printing settings.

With

$$\frac{\alpha \cdot e^{-\beta Z_{min}}}{-\beta} + K = C_{min} \tag{3}$$

and

$$\frac{\alpha \cdot e^{-\beta Z_{max}}}{-\beta} + K = C_{max}, \tag{4}$$

we get

$$\alpha = \frac{(-C_{max} - K) \cdot \ln\left(\frac{C_{max}-K}{C_{min}-K}\right) \cdot \left(\frac{C_{max}-K}{C_{min}-K}\right)^{\frac{Z_{max}}{Z_{min}-Z_{max}}}}{Z_{min} - Z_{max}} \tag{5}$$

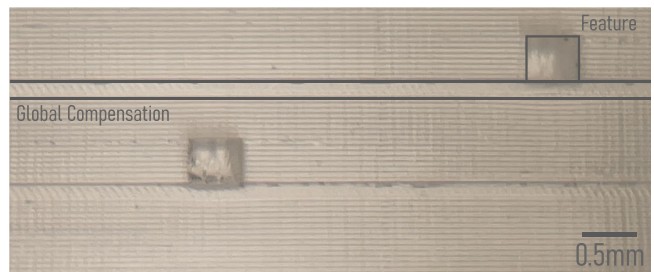

**Fig. 9 | Global compensation method of Flui3d.** A microscopic photograph from the side of a demonstrative 3D-printed microfluidic device utilizing global compensation with a slicing thickness of 50 μm.

and

$$\beta = \frac{ln\left(\frac{C_{max} - K}{C_{min} - K}\right)}{Z_{min} - Z_{max}}, for \frac{C_{min}}{C_{max}} \geq 0. \tag{6}$$

By putting $K = C_{min} + C_{max}$ into Eqs. (5) and (6) we get

$$\alpha = \frac{C_{min} \cdot ln\left(\frac{C_{min}}{C_{max}}\right) \cdot \left(\frac{C_{min}}{C_{max}}\right)^{\frac{Z_{max}}{Z_{min} - Z_{max}}}}{Z_{min} - Z_{max}} \tag{7}$$

and

$$\beta = \frac{ln\left(\frac{C_{min}}{C_{max}}\right)}{Z_{min} - Z_{max}}. \tag{8}$$

And finally, putting Eqs. (7) and Eq. (8) back to Eq. (2) we get the approximation formula for the dynamic height compensation calculation, which is used in Flui3d:

$$C(z) = -C_{min} \cdot \left(\frac{C_{max}}{C_{min}}\right)^{\frac{z}{Z_{min} - Z_{max}}} \cdot \left(\frac{C_{min}}{C_{max}}\right)^{\frac{Z_{max}}{Z_{min} - Z_{max}}} + C_{min} + C_{max} \tag{9}$$

In order to achieve optimal numbers for local compensation settings ($C_{min}@Z_{min}$ and $C_{max}@Z_{max}$) that are tailored to the specific printer model, print settings, and ambient factors (such as light source power, exposure time, resin properties, temperature, etc.), we offer a reference design model for users to acquire the appropriate settings. By printing this reference design and inspecting the resulting features, users can determine the extent of compensation required to achieve the desired level of precision and accuracy. This allows users to fine-tune their settings and achieve the best possible results with their specific setup. This reference design model and its use instructions are provided in Supplementary Data and Supplementary Note 6.

The global compensation function offers a further enhancement by adding a blank height after each feature layer (Fig. 9). This technique effectively minimizes unintended area curing by increasing the path through which light passes. Global compensation proves especially beneficial in situations where a design comprises multiple layers and local compensation proves insufficient or when users employ low-power light source 3D printers (i.e., require more exposure time per layer), such as entry-level LCD printers.

To optimize its effectiveness, it is crucial to carefully adjust the compensation height. Setting it too large may cause the printed design to become unstable. This adjustment depends on a variety of factors previously discussed. In cases where large local compensation values lead to layer cross-over, global compensation can help mitigate the issue. Similarly, when using printers with longer exposure times per layer, global

compensation can provide additional support to prevent curing in unintended areas. By incorporating extra space through global compensation, users can enhance the overall compensation process and avert layer cross-over.

The DFM helps users create print files for microfluidic designs that are optimized for the commonly used SLA manufacturing process and that can be easily and cost-effectively fabricated using 3D printing technology. These features enable users to efficiently optimize their output of designs for manufacturability, reliability, and cost and to avoid design errors and manufacturing defects.

## Data availability
The code for Flui3 is publicly available at https://github.com/TUM-EDA/Flui3d. All design files, detailed breakdown, and comparison of design complexities mentioned in this paper are provided in Supplementary Information section.

## Code availability
The code for Flui3d is publicly available at https://github.com/TUM-EDA/Flui3d.

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

## Acknowledgements

The authors would like to thank Xianer Chen, Xiaolin Ma, and Shen Hu, who gave help to the development process of Flui3d.

## Author contributions

Y.Z. devised the project, the main conceptual ideas and proof outline, carried out the implementation, conducted the experiments, derived the mathematical models, drafted the manuscript and designed the figures, M.L. commented on the manuscript, T.T. was involved in planning the experiments and contributed to the interpretation of the results, T.T. and U.S. supervised the project and edited the manuscript. All authors reviewed the manuscript.

## Funding

## Competing interests

The authors declare no competing interests.
