## [Peer Review File · Communications Engineering]

Reviewers' comments:

Reviewer #1 (Remarks to the Author):

The manuscript authored by Yushen Zhang et al., entitled “Open-Source Interactive Design Platform (Flui3d) for 3D-Printed Microfluidic Devices,” demonstrated the development of an open-source software package for the standard and customized design of 3D printable microfluidic devices. This new design platform features a few functions that are absent from existing 3D modeling software, particularly Manufacturing Optimization and Design Rule Check. Considering the importance of microfluidic devices for a vast range of research domains and the dominant role 3D printing will play for the fabrication, this work marks a significant advance towards the automated design and fabrication of microfluidics. The authors have conducted detailed introduction to the design rationale and protocols of operation. Therefore, the reviewer believe that it can be accepted for publication with minor modifications. Specifically,

1. 3D printing is a generic term for multiple additive manufacturing techniques. The manuscript mainly focuses on the effective design of SLA printable devices. Does the design fit all 3D printing mechanism, such as extrusion printing, laser direction writing, etc.? To provide a more informative publication, this end-to-end information is critical. It is suggested that either more analytical information is provided or the software is employed to print some parts with different 3D printing methods.
2. The compensation mechanisms introduced in the Design-for-Manufacturing might be the most useful feature for quality control and scaling down in size. However, quantitative metrics demonstrating the characteristics or performance of the DFM function is currently missing. Experimental data will be necessary to demonstrate the compensation effects under varying printing conditions.

Reviewer #2 (Remarks to the Author):

In this paper, Zhang et al. introduced an interactive web-based platform for designing 3D-printed microfluidic devices. The core idea is to incorporate some pre-defined components or modules into the platform to reduce the design complexity. While this platform may gain interest from microfluidic designers who are not familiar with traditional CAD tools, the following concerns need to be addressed before the paper can be considered for publication.

The major concern is how much flexibility the predefined components can provide to meet the versatile requirements of the practical applications. The current version only contains a limited number of components, and the parameters for tweaking them are also limited. Channels with non-rectangular cross-sections, such as circular shapes, can sometimes be useful in some specific applications. Is it possible to design these types of channels using the platform? And what DMF functions should be taken into account?

In addition to introducing the features and demos of the platform, the authors should provide more

information about the software architecture and design.

In the Design-for-Manufacturing (DFM) section, the authors should provide more details about what design rules and constraints (DRC) have been incorporated into the platform. And can show some examples to demonstrate the function better.

The authors only tested one 3D printer and one resin. More 3D printers and printing materials need to be tested to verify the availability of the platform, especially for DFM function.

Reviewer #3 (Remarks to the Author):

In this work, the authors reported an interactive software platform for designing microfluidic devices for 3D printing, which can offer a convenient solution for designing 3D models of microfluidic devices. The article is in details, but there are still several issues that need to be addressed before it can be recommended to be published in Communications Engineering.

1. In Introduction, the authors mentioned that “Despite the advantages of 3D printing for microfluidics, challenges remain in producing small and multi-layer 3D-printed microfluidics due to these limitations.” The authors should discuss the challenges details such as the limitation of printing accuracy and printing area, multi-materials and so on.
2. When the authors compare their results with other design tools in Figure 1, authors need to list more features such as livelink tools, simulations and so on.
3. The authors can discuss more about the advantages of Flui3D in designing process while comparing with others and list some examples. Such as the advantages comparing with Solidworks, Auto CAD, etc.
4. The cost of time and CPU should also be compared with traditional methods.
5. Font sizes are relatively small in some Figures.
6. More microfluidic devices can be utilized to validate the practicality of Flui3D.
7. Some grammar mistakes and spelling errors need to be corrected.
8. Some recent advances in 3D printing can be considered for the introduction to give a more comprehensive background i.e. Research 2020, 1426078., Small 2023, 2300047.

Dear reviewers,

Thank you for taking the time to carefully review our manuscript, "Open-Source Interactive Design Platform (Flui3d) for 3D-Printed Microfluidic Devices". We appreciate your valuable feedback and are grateful for the opportunity to address your concerns.

We have carefully reviewed your comments and have made revisions to the manuscript. Please allow us to respond to your comments and clarify the revisions we have made. Our point-by-point response to your comments and all the revision changes are highlighted here and in our revised manuscript in blue for your convenience.

Reviewer 1:

The manuscript authored by Yushen Zhang et al., entitled "Open-Source Interactive Design Platform (Flui3d) for 3D-Printed Microfluidic Devices," demonstrated the development of an open-source software package for the standard and customized design of 3D printable microfluidic devices. This new design platform features a few functions that are absent from existing 3D modeling software, particularly Manufacturing Optimization and Design Rule Check. Considering the importance of microfluidic devices for a vast range of research domains and the dominant role 3D printing will play for the fabrication, this work marks a significant advance towards the automated design and fabrication of microfluidics. The authors have conducted detailed introduction to the design rationale and protocols of operation. Therefore, the reviewer believe that is can be accepted for publication with minor modifications. Specifically,

1. 3D printing is a generic term for multiple additive manufacturing techniques. The manuscript mainly focuses on the effective design of SLA printable devices. Does the design fit all 3D printing mechanism, such as extrusion printing, laser direction writing, etc.? To provide a more informative publication, this end-to-end information is critical. It is suggested that either more analytical information is provided or the software is employed to print some parts with different 3D printing methods.

Firstly, we sincerely appreciate your insightful comments and invaluable suggestions. We acknowledge that 3D printing encompasses various additive manufacturing techniques. To offer a more comprehensive understanding, we have clarified in the Introduction section on page 2 that, although technologies like FDM or PolyJet printing may be employed for microfluidics fabrication, SLA currently stands out as the most widely used technology for 3D printed microfluidics due to its superior accuracy and precision. Nevertheless, our design platform generates STL files as output, a standard format for 3D printing. This means that all printing technologies, including extrusion printing and laser direction writing, can benefit from the design convenience provided by Flui3d.

In response to your recommendation for more analytical information or employing software to print parts with different 3D printing methods, we have expanded our experimental setup by acquiring an entry-level home-use MSLA 3D printer and using two

different resins: a home-use, cost-effective, non-professional hobby resin, Anycubic Plant Based + Clear, and a professional biocompatible resin, Liqcreate Bio-Med Clear. Our experiments successfully demonstrated the printing of a microfluidic device designed with Flui3d with both the DLP and MSLA printers, using each resin. The detailed results are added in our supplementary information and mentioned this in the Results section of our revised manuscript on pages 6-8.

2. The compensation mechanisms introduced in the Design-for-Manufacturing might be the most useful feature for quality control and scaling down in size. However, quantitative metrics demonstrating the characteristics or performance of the DFM function is currently missing. Experimental data will be necessary to demonstrate the compensation effects under varying printing conditions.

Thank you very much for your valuable feedback. We have addressed the concern about the lack of quantitative metrics for the Design-for-Manufacturing (DFM) function. We have acquired a new printer featuring another printing technology compared to the DLP printer mentioned previously in our manuscript. Additionally, we have obtained two new resins with different properties, namely a plant-based resin and a biocompatible resin. We printed the three-layer, three-way mixing device mentioned in our manuscript using both printers. This involved utilizing the original resin (Miicraft BV-007A) mentioned earlier, as well as the newly acquired two resins. Notably, we incorporated our Design for Manufacturability (DFM) functionality from Flui3d to showcase its efficacy across various printing technologies and resin types. In our revised manuscript, we have included additional supplementary information (in Additional Validation Experiment section of "Print.pdf"), complete with DFM settings for these varying printing conditions.

Reviewer 2:

In this paper, Zhang et al. introduced an interactive web-based platform for designing 3D-printed microfluidic devices. The core idea is to incorporate some pre-defined components or modules into the platform to reduce the design complexity. While this platform may gain interest from microfluidic designers who are not familiar with traditional CAD tools, the following concerns need to be addressed before the paper can be considered for publication.

The major concern is how much flexibility the predefined components can provide to meet the versatile requirements of the practical applications. The current version only contains a limited number of components, and the parameters for tweaking them are also limited. Channels with non-rectangular cross-sections, such as circular shapes, can sometimes be useful in some specific applications. Is it possible to design these types of channels using the platform? And what DMF functions should be taken into account?

Thank you for raising this question. In the current version of Flui3d, we have tried to define as many standardized components as possible, including:

- Chamber: width, length, height, and corner radius.
- Chamber with pillars: width, length, height, corner radius, number of pillar rows,

- number of pillar columns, and pillar radius.
- Serpentine channel: channel width, length, height, channel distance, number of turnings, and corner radius.
- Tesla valve: channel width, height, number of segment pairs, segment length, and segment width.
- Droplet generator: disperse channel width, continuous channel width, droplet channel width, droplet channel length, connection channel length, and height.
- Channel width transition: transition length, transition width left, transition width right, connection channel length, and height.
- Channel height transition: transition length, transition height left, transition height right, connection channel length, and height.

And for general components:

- Channel: width, length, and height.
- Port: shape, port radius, chamfer radius, and chamfer depth.
- Bridge: height, degree of left slope, degree of right slope, and radius.
- Channel bending corner: radius.
- Via: pass through layers, shape, and radius.

Flui3d also provides a custom component design function. We encourage users to share and use custom components in their own designs, increasing the range of available components and fostering collaboration and innovation. In the future, we will add additional components based on users' feedback and sharing into Flui3d's component library.

Furthermore, thank you very much for your second question. It is a valuable suggestion. Indeed, circular channels are useful in some specific applications. We will consider supporting circular channels with associated DFM functions in our future version.

In addition to introducing the features and demos of the platform, the authors should provide more information about the software architecture and design.

We totally agree with this comment. We have added additional information regarding the software design and architecture in the revised manuscript's Methods section on page 15 and in the supplementary information section (in "SoftwareDesign.pdf").

In the Design-for-Manufacturing (DFM) section, the authors should provide more details about what design rules and constraints (DRC) have been incorporated into the platform. And can show some examples to demonstrate the function better.

Thank you very much for this valuable comment. We have added information about our DRC and the rules that are incorporated in the Methods section of the revised manuscript on page 19.

The authors only tested one 3D printer and one resin. More 3D printers and printing materials need to be tested to verify the availability of the platform, especially for DFM function.

We sincerely appreciate your invaluable comment. We totally agree with it. Apart from the DLP printer and the Miicraft BV-007A resin mentioned in our original manuscript, we now have expanded our experimental setup by buying an entry-level home-use MSLA 3D printer, the Elegoo Mars 4, and using two different new resins: a home-use, cost-effective, non-professional hobby resin, Anycubic Plant Based + Clear, and a professional biocompatible resin, Liqcreate Bio-Med Clear. We incorporated our Design for Manufacturability (DFM) functionality from Flui3d to showcase its efficacy across various printing technologies and resin types. In our revised manuscript, we have included supplementary information, complete with DFM settings used in our newly added experiments, and mentioned this in the Results section of our revised manuscript on pages 6-8. In the Additional Validation Experiment section of the supplementary information's Print.pdf file, we have showcased the fabricated chips of the three-layer, three-way mixer design.

Reviewer 3:

In this work, the authors reported an interactive software platform for designing microfluidic devices for 3D printing, which can offer a convenient solution for designing 3D models of microfluidic devices. The article is in details, but there are still several issues that need to be addressed before it can be recommended to be published in Communications Engineering.

1. In Introduction, the authors mentioned that “Despite the advantages of 3D printing for microfluidics, challenges remain in producing small and multi-layer 3D-printed microfluidics due to these limitations.” The authors should discuss the challenges details such as the limitation of printing accuracy and printing area, multi-materials and so on.

The authors sincerely appreciate your insightful comments and suggestions. We totally agree with this and have now added more information on the limitations and challenges in 3D-printed microfluidics in the Introduction section on page 2.

2. When the authors compare their results with other design tools in Figure 1, authors need to list more features such as livelink tools, simulations and so on.

Thank you very much for your suggestion. We totally agree and have added additional information and comparison in Figure 1.

3. The authors can discuss more about the advantages of Flui3D in designing process while comparing with others and list some examples. Such as the advantages comparing with Solidworks, Auto CAD, etc.

Thank you very much for your suggestion. We have added additional information in the introduction and highlighted Flui3d's advantages over commercial tools in the discussion section on pages 13-14.

4. The cost of time and CPU should also be compared with traditional methods.

Thank you for your insightful comment. We have described the design complexity, which is in correlation with design time costs, in the Results section under the heading "Design Complexity." For a more detailed breakdown and comparison, please refer to the Supplementary Information section (in "Complexities.pdf").

The cost of the CPU is depending on the user's hardware and, therefore, may vary a lot. Generally, when comparing CPU costs, Flui3d stands out as a web application that operates exclusively through a web browser, providing a significantly higher efficiency for clients compared to traditional software requiring local installation. For instance, exporting an STL file using an Apple MacBook Pro 2020 (M1/16GB) with Flui3d through Safari browser incurs a CPU usage of 1.3% and RAM usage of 70.9MB. In contrast, the same operation with AutoCAD for Mac 2024 demands a CPU usage of 28.7% and a RAM usage of 1.3GB.

5. Font sizes are relatively small in some Figures.

Thank you very much for this comment. We have adjusted the font size in Figure 6 and Figure 8.

6. More microfluidic devices can be utilized to validate the practicality of Flui3D.

Thank you for your suggestions. We have made Flui3d online accessible, and we encourage everyone to try to validate the design process of Flui3d and share their designs across the community. We believe that with the help of the community, we can strengthen and optimize Flui3d further. Additionally, we have bought an additional entry-level printer using MSLA printing technology and two additional resins with different properties. We successfully validated the printability of using Flui3d designed microfluidic device with both the DLP printer mentioned previously in our original manuscript and the newly acquired MSLA printer using different resins. In our revised manuscript, we have included supplementary information ("Print.pdf") regarding our newly added experiments.

7. Some grammar mistakes and spelling errors need to be corrected.

Thank you very much for this comment. We have checked and corrected misspellings and grammatical errors in our revised manuscript with the help of a professional tool.

8. Some recent advances in 3D printing can be considered for the introduction to give a more comprehensive background i.e. Research 2020, 1426078., Small 2023, 2300047.

We totally agree. Thank you very much for sharing those insightful articles. We have added to our introduction on page 2 more comprehensive background information about recent advances in 3D printing technology.

We look forward to hearing from you regarding our submission and to respond to any further questions and comments you may have.

Sincerely,
Tsun-Ming Tseng

January 29, 2024

REVIEWERS' COMMENTS:

Reviewer #1 (Remarks to the Author):

The authors have developed a versatile and useful software for the design of 3D printable microfluidic devices mainly with SLA methods. The revised article has provided additional information and data that addressed my previous questions. Therefore, it is suggested for publication on Communications Engineering.

Reviewer #2 (Remarks to the Author):

All the comments have been addressed properly.

Reviewer #3 (Remarks to the Author):

The paper can be accepted without further revision.

Dear reviewers,

Thank you for taking the time to carefully review our manuscript, "Open-Source Interactive Design Platform for 3D-Printed Microfluidic Devices". Again, we appreciate your valuable feedback and giving us the opportunity to improve our manuscript.

Reviewer #1 (Remarks to the Author):

The authors have developed a versatile and useful software for the design of 3D printable microfluidic devices mainly with SLA methods. The revised article has provided additional information and data that addressed my previous questions. Therefore, it is suggested for publication on Communications Engineering.

Reviewer #2 (Remarks to the Author):

All the comments have been addressed properly.

Reviewer #3 (Remarks to the Author):

The paper can be accepted without further revision.

Sincerely,
Tsun-Ming Tseng

February 29, 2024